# Occurrence of major infectious diseases and healthcare seeking among young children with disabilities in Sierra Leone using cross-sectional population-based survey data

Anna-Theresia Ekman  ,[1] Esagila Cherry,[2] Paul Sengeh,[3] Nance Webber,[3] Mohammad Bailor Jalloh,[3] Nicola Orsini,[1] Tobias Alfvén,[1] Helena Frielingsdorf Lundqvist[2]

A-TE and EC are joint first authors.

[1]Global Public Health, Karolinska Institutet, Stockholm, Sweden
[2]Department of Health Medicine and Caring Sciences, Linkoping University, Linkoping, Sweden
[3]FOCUS1000, Freetown, Sierra Leone

**Correspondence to**
Dr Anna-Theresia Ekman; anna-theresia.ekman@ki.se

## ABSTRACT

**Background**  Children with disabilities are at risk of worse health outcomes compared to children without functional difficulties. Sierra Leone has one of the world's highest prevalences of functional difficulties among children, but little is known about the co-occurrence of major infectious diseases and healthcare-seeking behaviours among children with disabilities.

**Methods**  We used household survey cross-sectional data on children 2–4 years old and logistic regression models estimating ORs between functional difficulties and symptoms of infectious diseases including diarrhoea, fever and acute respiratory infection (ARI), adjusted for sex, age and stunting. We also examined whether caregivers sought advice or treatment for the illness from any source and if the child was given any treatment for the illness.

**Results**  There was an increased risk of fever among children with functional difficulty (adjusted OR (AOR)=1.3, 95% CI 1.1 to 1.8) and children with severe functional difficulty (AOR=1.6, 95% CI 1.0 to 2.7). Children with severe functional difficulty were also at increased risk of diarrhoea (AOR=1.8, 95% CI=1.1 to 3.3). There were no significant differences in seeking advice or treatment for diarrhoea, fever or ARI symptoms between the groups.

**Conclusions**  In Sierra Leone, children with functional difficulties, especially severe functional difficulties, more often have symptoms of major childhood diseases that are known to increase under-5 mortality.

## INTRODUCTION

Almost 240 million children worldwide have functional difficulties, with the highest prevalence in West Africa.[1] Persons with disability experience significantly poorer health than their non-disabled peers, ultimately risking a shortened lifespan.[2–5] To meet the 2030 Agenda pledge of leaving no one behind,[6] an increased understanding of health and healthcare-seeking behaviour among children with disabilities is needed.

---

### WHAT IS ALREADY KNOWN ON THIS TOPIC

⇒ Functional difficulty among young children is common in Sierra Leone. Even so, there is lacking information on major infectious diseases and healthcare-seeking among children with functional difficulties in Sierra Leone.

### WHAT THIS STUDY ADDS

⇒ We found that the odds of fever was 30% higher among children with functional difficulty (AOR=1.3, 95% CI 1.1 to 1.8), and 60% higher among children with severe functional difficulty (AOR=1.6, 95% CI 1.0 to 2.7). For children with severe functional difficulties, it was also more common with diarrhoea (AOR=1.8, 95% CI=1.1 to 3.3).

### HOW THIS STUDY MIGHT AFFECT RESEARCH, PRACTICE OR POLICY

⇒ To overcome these health inequities, efforts must be made to include children with disabilities in preventive and health promoting child health as well as curative care, while acknowledging that targeted interventions tailored to the needs of children with disabilities will be of additional benefit.

---

Five million children in the world still die before their fifth birthday, with pneumonia, diarrhoea and malaria being the leading causes of under-5 mortality after the neonatal period.[7] All conditions are largely preventable and treatable through cost-effective measures, and malnutrition is an important risk factor.[7]

In Sierra Leone, the under-5 mortality rate is 109 deaths per 1000 live births.[8] Malnutrition is widespread, with 30% of children below 5 being stunted, 5% being wasted and 14% being underweight in 2019.[9] Mothers and caretakers of young children who are suspected of having a disease normally seek

healthcare from sources such as community health providers. The lack of health information on children living with disabilities in Sierra Leone has previously been noted.[10]

In 2010, the Government introduced 'The Sierra Leone Free Healthcare Initiative' to ensure universal health coverage by providing free preventive and curative health services for pregnant women, lactating mothers and children under 5 years of age.[11] An updated version of the policy, issued post-Ebola, makes some provision for services to disabled persons, and such services will also cover drugs and consultations. However, the full extent of such services is not specified.

The child functioning module (CFM) was introduced during the 2017 Sierra Leone Multiple Indicator Cluster Survey (MICS), as a novel tool to collect data on functional difficulties among children. In line with the biopsychosocial model, the CFM collects data on functional difficulties rather than specific biomedical impairments. The underlying impairments captured by the tool are heterogeneous and include both mild-to-severe impairments. The 2017 Sierra Leone MICS also provides information on leading causes of under-5 mortality, measured through episodes of fever (proxy for malaria), acute respiratory infection (ARI) symptoms (proxy for pneumonia) and diarrhoea.[12]

Enhancing knowledge on the relationship between disability and leading causes of under-5 mortality can inform health planning policy to accurately reflect the needs of children, ultimately improving life length and quality for a vulnerable population. The aim of this article is therefore to examine the association between symptoms of infectious diseases and disability among young children in Sierra Leone, compared with children without disabilities. A secondary aim is to study healthcare-seeking behaviour and access to treatment for the same symptoms in the same population.

## METHODS

### Study design and description of the data set

This study uses two staged cluster-survey data from the 2017 Sierra Leone MICS.[13] The survey included 11 774 children under 5 of whom 11 764 mothers/caretakers were interviewed (response rate 99.9%).[14] Data on functional difficulties was collected from children 2–4 years and therefore our study population was limited to children within this age span, in total 7117 children. Sample allocation and participant selection have been described in detail.[14] Data were extracted from the dataset for children under 5, including information about functional difficulties from the CFM (figure 1), and information about disease episodes, care-seeking and access to treatment from the care of illness module (figure 2).

### Operationalising disability

The CFM estimates children at risk of disability through a Likert-like scale, collecting information on functional difficulties in the functional areas of seeing, hearing, mobility, dexterity, communication, learning, behaviour and playing, using primary caregiver (usually mothers) as proxy responders.[15] Children with functional difficulties were

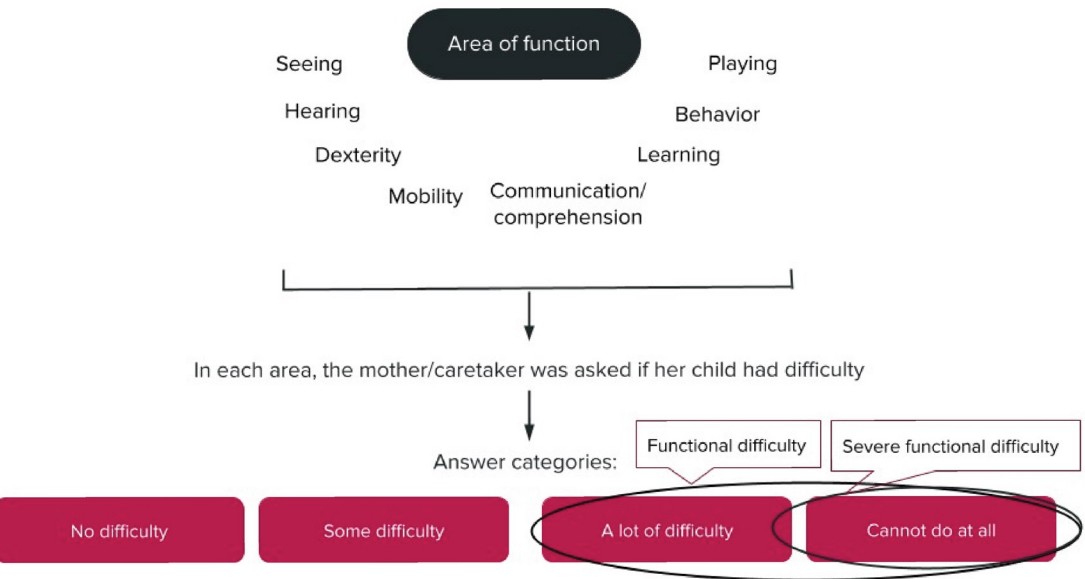

**Figure 1** Identifying children with functional difficulties. The child functioning module asked the mother/caretaker a set of questions on eight areas of functioning regarding their child. Each area was assessed against a rating scale, with the answer categories 'no difficulty', 'some difficulty', 'a lot of difficulty' and 'cannot do at all'. The child was considered having a functional difficulty at the response of 'a lot of difficulty' or 'cannot do at all', and severe functional difficulty at the response of 'cannot do at all'. The only exception was the domain of controlling behaviour, where the response categories were instead 'not at all', 'less', 'the same', 'more' and 'a lot more' (not presented in the figure). The child was considered having a functional difficulty at the response of 'a lot more'.

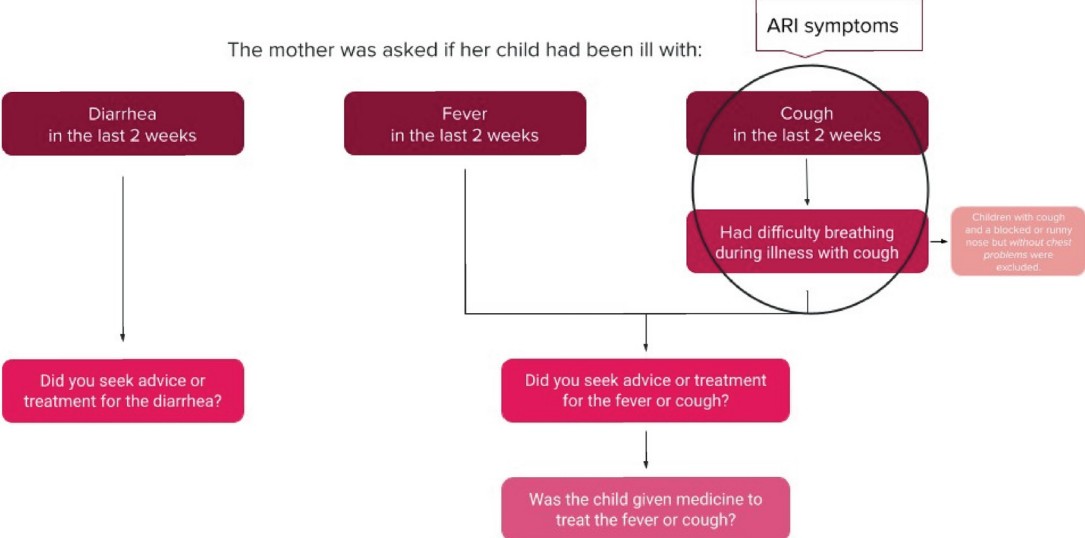

**Figure 2** Identifying episodes of illness, healthcare seeking and access to treatment. In the survey's care of illness module, the mother/caretaker was asked if the child had been ill with diarrhoea, fever or cough in the last 2 weeks. If the child had been ill with cough, the mother/caretaker was asked if the child had difficulty breathing during that illness. The child was considered having ARI symptoms if it had cough and difficulty breathing at the same time, meaning cough due to a chest problem. To identify healthcare-seeking behaviour, the mother/caretaker was asked if they had sought advice or treatment for the illnesses, respectively. To identify access to treatment, the mother/caretaker was asked if the child had been given medicine for the fever or cough. ARI, acute respiratory infection

defined as those with a lot of difficulties and/or who cannot function at all in one or more of the different domains of functioning (figure 1). A child with severe functional difficulty was defined as 'cannot do at all' in any of the domains. The only exception was the domain of controlling behaviour, where a child was considered having a functional difficulty at the response of 'a lot more'.

### Episodes of diarrhoea, fever or respiratory difficulties

The caregiver was asked to estimate whether the child had an episode of diarrhoea, fever or ARI symptoms during the preceding two-week period. These survey questions have previously been used in analyses of MICS surveys.[16 17] ARI symptoms were defined as rapid or difficult breathing during illness with cough, when symptoms were perceived to be due to a problem in the chest or both a problem in the chest and a blocked or runny nose.[14 18] Thus, only having symptoms of an upper respiratory tract infection were not considered as ARI symptoms. If the child had either diarrhoea, fever or ARI symptoms, the caregiver was asked if they had sought any advice or treatment for the illness from any source and if the child was given any medicine to treat the illness (figure 2).

### Statistical analysis

Functional difficulty/severe functional difficulty were used as independent variables. Potential predictors considered were child age's, child's gender, stunting, household wealth, maternal level of education and rural–urban residence status. Crude (OR) and adjusted OR (AOR) with 95% CI were estimated with multivariable logistic regression models. Stata V.16.0 was used for all survey analyses with the stratum variable from the household dataset and centring single units.

### Public involvement

Due to this being a secondary data analysis, it was not possible to involve the public in the design, conduct, reporting or dissemination plans of our research.

### RESULTS

The weighted sample included 7090 children age 2–4 years, of which 7% were identified as having functional difficulties and almost 2% were identified as having severe functional difficulties (table 1). One in five children (20%) had fever in the last 2 weeks, 7% had diarrhoea and 2% had ARI symptoms. Functional difficulty was associated with child's age, child's gender and stunting, and these factors were included in the adjusted logistic regression. The odds of fever was 30% higher among children with functional difficulty (AOR=1.3, 95% CI 1.1 to 1.8) (table 2). The odds of fever was 60% higher among children with severe functional difficulty (AOR=1.6, 95% CI 1.0 to 2.7). For children with severe functional difficulties, it was also significantly more common with diarrhoea (AOR=1.8, 95% CI=1.1 to 3.3). There was no significant difference for ARI symptoms after adjusting for confounding.

For more than two-thirds of the children, caregivers had sought advice or treatment, and almost all were given some form of medicine for fever or cough. There were no significant differences in seeking advice or treatment for diarrhoea, fever or ARI symptoms (table 3). However,

Table 1  Weighted sample characteristics of the study population 2–4 year olds

| Variable | Total, N (%) |
|---|---|
| Total number of children | 7090 (100%) |
| Children with functional difficulties | 471 (7%) |
| Children with severe functional difficulties | 126 (2%) |
| Child factors | |
| Sex | |
| Male | 3504 (59) |
| Female | 3586 (51) |
| Age | |
| 2 years | 2388 (34) |
| 3 years | 2351 (33) |
| 4 years | 2351(33) |
| Malnutrition | |
| Underweight | 760 (11) |
| Stunted | 2121 (30) |
| Wasted | 221 (3) |
| Caregiver factors | |
| Education level | |
| Preprimary or none | 4528 (64) |
| Primary | 853 (12) |
| Junior secondary | 875 (12) |
| Senior secondary or higher | 834 (12) |
| Household and environmental factors | |
| Area | |
| Urban | 2663 (38) |
| Rural | 4426 (62) |
| Wealth | |
| Poorest | 1679 (24) |
| Second | 1595 (23) |
| Middle | 1482 (21) |
| Fourth | 1222 (17) |
| Richest | 1112 (16) |
| Common childhood illnesses | |
| Occurrence of illness during the preceding 2 weeks | |
| Diarrhoea | 480 (7) |
| Fever | 1460 (21) |
| Cough | 1157 (16) |
| Difficulty breathing during illness with cough | 452 (6) |
| Acute respiratory infection | 111 (2) |
| Healthcare-seeking behaviour | |
| Sought advice or treatment for diarrhoea from any source | 336 (70) |
| Sought advice or treatment for the fever or acute respiratory infection | 1070 (72) |

Continued

Table 1  Continued

| Variable | Total, N (%) |
|---|---|
| Access to treatment | |
| Given any type of treatment for diarrhoea | 414 (86) |
| Given medicine to treat the fever or acute respiratory infection | 1348 (91) |

there was a trend towards less healthcare seeking for diarrhoea in children with functional difficulties and severe difficulties, and a trend towards more healthcare seeking for fever or ARI symptoms (table 3). In this material, there were no differences between children with or without functional difficulties in odds of being given medicine to treat the illness.

## DISCUSSION

In this study, children with functional difficulties were more prone to experience episodes of fever, and children with severe functional difficulties were significantly more likely to be ill with diarrhoea, compared with children without functional difficulty. We found no significant difference on seeking or accessing treatment between children with or without functional difficulties.

Several conditions leading to functional difficulties, such as autism spectrum disorder, attention deficit hyperactivity disorder, cerebral palsy and seizures/epilepsy, are associated with somatic, psychiatric and neurological comorbidities.[19] For example, children with cerebral palsy often have dysphagia, which places them at risk for aspiration pneumonia.[20] This may be part of the reason behind the association between functional difficulty and the examined symptoms and infectious diseases.

Even though it is not possible to understand the underlying impairments, it is likely that children with severe functional difficulties have more outspoken conditions, such as severe cerebral palsy. Indeed, a recent article investigating the same study population found a high risk of comorbidity between difficulties in different functional domains,[21] which reflects that some children within the group are more affected than others.

Another possible mechanism of association may be malnutrition. The relationship between malnutrition and childhood disability has been confirmed in multiple studies,[22] and malnutrition is a shared risk factor for pneumonia, diarrhoea and malaria.[23] It has also been established that children with functional difficulties in Sierra Leone have an increased level of stunting at 38%.[21] While this could explain part of the observed relationship between disability and infectious diseases, the increased odds of symptoms of infectious diseases persist after adjusting for stunting in this study population. The plethora of biomedical conditions captured by the CFM makes it difficult to fully entangle the relationship among disability, infectious diseases and malnutrition.

**Table 2** Weighted ORs of infectious diseases in children with and without functional difficulty

| | Functional difficulties | | Severe functional difficulties | |
|---|---|---|---|---|
| | OR (95% CI) | Adjusted* OR (95% CI) | OR (95% CI) | Adjusted* OR (95% CI) |
| Occurrence of illness during the preceding 2 weeks | | | | |
| Diarrhoea | 1.4 (1 to 2) | 1.3 (0.9 to 1.8) | 2 (1.1 to 3.6) | 1.8 (1 to 3.3) |
| Fever | 1.4 (1 to 1.9) | 1.3 (1 to 1.8) | 1.7 (1 to 2.7) | 1.6 (1 to 2.7) |
| Cough | 1 (0.8 to 1.4) | 0.9 (0.7 to 1.3) | 2 (1.2 to 3.5) | 2 (1.1 to 3.3) |
| Difficulty breathing during illness with cough | 1.4 (1 to 2) | 1.3 (0.9 to 2) | 1.7 (0.8 to 3.8) | 1.7 (0.7 to 3.7) |
| Acute respiratory infection | 1.8 (1 to 3.5) | 1.7 (0.9 to 3.3) | 2.6 (0.9 to 7.2) | 2.4 (0.8 to 7) |

*Adjusted for child's age, sex and stunting.

The increased susceptibility to major infectious diseases among children with disability has previously been described. A multilateral analysis of 3–4 year olds at risk for intellectual disability using MICS data from 2010 to 2015 and data from 24 low-income and middle-income countries found an increased risk of symptoms of ARI, diarrhoea and fever.[17] Although it did not comment on country-specific results, estimates for Sierra Leone were reported, showing that children at-risk of intellectual disability were significantly less likely to have diarrhoea.[17] Our results align with the study's pooled findings for low-income countries, but differ from the study's findings on Sierra Leone.

A secondary finding is that one in five children in Sierra Leone had a fever during the previous 2 weeks, likely reflecting that fever is an imprecise proxy for malaria. The high proportion in this survey is in line with results from similar surveys.[16]

### Strengths and limitations

This study builds on a large and nationally representative study population, and the introduction of the CFM may allow for comparison between settings and to repeat measures in similar study populations over time. However, cross-sectional data cannot be used to draw conclusions about causality, and it is not possible to know what types of impairments underpin the individual child's functional difficulties.

The proxies used for the infectious illnesses may have low specificity; it has been described that not all reported ARI symptoms are in fact true pneumonia[18] and fever can be caused by other diseases than malaria. In some surveys, a rapid diagnostic test for malaria is included, but this was not the case in the Sierra Leone 2017 MICS. However, all disease proxies used are well established in household surveys. Also, the fact that caregivers report symptoms from the two weeks preceding the survey, makes it likely that estimated prevalences are susceptible to seasonal changes in disease patterns. This study focuses on the difference between groups of children which we believe are less affected by seasonal variation.

We could not document significant differences in healthcare-seeking behaviour or access to treatment. While this may be true, it can also be due to the relatively small number of children with functional difficulties in the dataset who had symptoms of each infectious disease. This was also the reason why the multilateral analysis mentioned above could not generate country-specific estimates of access to treatments.[17]

**Table 3** Weighted ORs of healthcare-seeking behaviour in children with and without functional difficulty

| | Functional difficulties | | Severe functional difficulties | |
|---|---|---|---|---|
| | OR (95% CI) | Adjusted* OR (95% CI) | OR (95% CI) | Adjusted* OR (95% CI) |
| Healthcare-seeking behaviour | | | | |
| Sought advice or treatment for diarrhoea from any source | 0.8 (0.4 to 1.6) | 0.8 (0.4 to 1.6) | 0.9 (0.3 to 3) | 0.9 (0.3 to 3.1) |
| Sought advice or treatment for the fever or acute respiratory infection | 1.2 (0.7 to 2.1) | 1.2 (0.7 to 2) | 1.3 (0.5 to 3.7) | 1.3 (0.5 to 3.8) |
| Access to treatment | | | | |
| Given any type of treatment for diarrhoea | 0.8 (0.3 to 1.9) | 1 (0.4 to 2.5) | 1 (0.2 to 5) | 1.3 (0.2 to 6.6) |
| Given medicine to treat the fever or acute respiratory infection | 1 (0.5 to 2.2) | 1.1 (0.6 to 2.4) | 1.1 (0.3 to 4.3) | 1.3 (0.3 to 4.7) |

*Adjusted for child's age, sex and stunting.

## CONCLUSION

In Sierra Leone, children with functional difficulties, especially severe functional difficulties, are more affected by major childhood diseases that are known to increase under-5 mortality. To overcome these health inequities, effort must be made to include children with disabilities in preventive and health promoting child health as well as curative care, while acknowledging that targeted interventions tailored to the needs of children with disabilities will be of additional benefit. Investing in health equity could result in lowered child morbidity and mortality, ultimately fulfilling the 2030 Agenda's pledge of leaving no one behind.

**Contributors** A-TE and EC contributed equally to the work. A-TE is responsible for the overall content as guarantor. A-TE, EC and HFL conceptualised the idea and designed the study. PS, NW, MBJ, NO and TA provided input on the proposed research questions as well as methodology. All authors contributed to the subsequent drafts. All authors approved the final manuscript as submitted and agreed to be accountable for all aspects of the work.

**Funding** This work was supported by Swedish regional ALF funding 2019. ALF is the abbreviation in Swedish of an agreement between the central government and seven regions on physician education and clinical research.

**Competing interests** HFL has received ALF funding for other studies in 2022.

**Patient and public involvement** Patients and/or the public were not involved in the design, or conduct, or reporting, or dissemination plans of this research.

**Patient consent for publication** Not applicable.

**Ethics approval** This study involves human participants. Ethical approval for data collection was attained from the Ethics and Scientific Review Committee in Sierra Leone by the Sierra Leone MICS team. Informed consent was obtained from all participants during data collection, and data have been deidentified for confidentiality. This study is a secondary analysis of the anonymised data from MICS, publicly accessible for research purposes. There is no number/ID to report for this secondary data analysis, as the ethical approval was sought by the Sierra Leone MICS team and reported in the official Sierra Leone MICS 2017 report. Participants gave informed consent to participate in the study before taking part.

**Provenance and peer review** Not commissioned; externally peer reviewed.

**Data availability statement** Data are available in a public, open access repository. Data are available through the UNICEF multiple indicator cluster survey webpage (https://mics.unicef.org/).

**ORCID iD**
Anna-Theresia Ekman http://orcid.org/0000-0003-0647-8817

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
