## [Reviewer comments · BMJ Paediatrics Open]

ARTICLE DETAILS

TITLE (PROVISIONAL)	The occurrence of major infectious diseases and health care seeking among young children with disabilities in Sierra Leone using cross-sectional population-based survey data
AUTHORS	Ekman, Anna-Theresia Cherry, Esagila Sengeh, Paul Webber, Nance Jalloh, Mohammad Orsini, Nicola Alfvén, Tobias Frielingsdorf Lundqvist, Helena

VERSION 1 – REVIEW

REVIEWER	Dr. Oluwatosin Eunice Olorunmoteni Obafemi Awolowo University Department of Paediatrics and Child Health Obafemi Awolowo University/ Obafemi Awolowo University Off Ede Road Ile-Ife Ile-Ife Osun 220005 Nigeria
REVIEW RETURNED	07-Mar-2024

GENERAL COMMENTS	Dear Authors, Thank you for this important work on the occurrence of infectious diseases and health-seeking behaviour of children with disabilities in Sierra Leone. The study brings to the fore the double burden of childhood diseases. Abstract: The abstract section reads thus: "Sierra Leone has one of the world's highest prevalence of functional difficulties among children". Kindly change worlds in this sentence to world's. Introduction: The introduction section also has a few spelling errors. Hence, I advise the authors to use an English language grammar checker to assist in improving the English language. Kindly provide information on the structure of the health system in Sierra Leone, stating how much support is available for people with disabilities. Also, is the cost for the treatment of ARI out of pocket or is there insurance coverage? These will help to understand the health-seeking behaviour of the respondents. Methods: Why was the study population limited to children aged 2-4 years? The symptoms being studied for possible association with functional difficulty are the symptoms of the top diseases causing under-five mortality. Why is the study population not under-five children? Results: Kindly edit the following: The odds of having fever were about one-fold higher among children with functional difficulty (aOR= 1.3, 95% ci 1.1 - 1.8). The odds of having fever were about twofold higher among children with severe functional difficulty (aOR = 1.6, 95% ci 1.0-2.7)... etc.
--

REVIEWER	Dr. Andrew Williams Virtual Academic Unit Virtual Academic Unit, Childrens Directorate Virtual Academic Unit, Childrens Directorate Northampton General Hospital Northampton NN1 5BD United Kingdom of Great Britain and Northern Ireland
REVIEW RETURNED	11-Mar-2024
GENERAL COMMENTS	I am really impressed with this paper and recommend publication. It makes a clear, important contribution to the literature of child healthcare and most particularly for young children with suspected or diagnosed functional/ severe functional disability in a country which has many significant ongoing challenges.. I congratulate this paper's authors. As there is no place where I can write directly to the journal editor for the following reason, I am obliged to put my additional comments here. I believe that there is an exciting second paper - an editorial to be written ? BMJ ? Archives of Disease of Childhood. That paper would enlighten Western child health practitioners not only to the impressive nature of this already submitted paper but also of the Focus 1000, with its interventions in child health and extensive Kombra Network. https://focus1000.org/ The authors have produced a very strong paper which is powerful with the numbers of patients recruited. I was particularly struck by its wider context. Firstly, the '2017 Sierra Leone MICS. The survey included 11,774 children under five of whom 11,764 mothers/caretakers were interviewed (response rate 99.9%) Secondly, that this hugely impressive figure is achieved within a population where 64% of caregivers had either pre primary or no education and only 12% had senior secondary education or higher. The three Focus 1000 co authors should be the authors of this second paper and I would hope that they are commissioned to write it.

VERSION 1 – AUTHOR RESPONSE

Dear Dr Olusanya and Dr Raman,

Thank you for highlighting this. There are no grant numbers for regional ALF grants. However, to contribute to transparency, the title of the grant application was "Global psykisk hälsa: Multiple indicator cluster survey (MICS)-studie av prevalens och riskfaktorer för psykisk funktionsnedsättning bland barn och ungdomar i Sierra Leone."

All figures have been removed and are instead attached in line with the recommended format and resolution demands.

Thank you for finding the time to read and to provide input on our manuscript! A detailed response is provided below.

We are thankful that you highlighted this error. It has been corrected to world's.

The manuscript has been revised using an English language grammar checker set on British English. We have added the proposed background information on the Sierra Leone health system starting in the third paragraph of the introduction.

The child functioning module used only provide information on children 2-4 years, making it impossible to widening the study population using this method. This has been clarified in the first paragraph of the methodology section.

Thank you for this suggestion. We suggest keeping the percentage statement, as it makes the differences clearer to the reader and is a bit more nuanced.

Dr. Andrew Williams, Virtual Academic Unit

Thank you for finding to time to read and provide input on our manuscript! We are also continuously impressed with the collection of data through these surveys and the ongoing work for children with disabilities. We wholeheartedly support the writing of an editorial or similar by the authors working at FOCUS1000, to share its experience with a wider audience!

VERSION 2 – REVIEW

REVIEWER	Dr. Oluwatosin Eunice Olorunmoteni Obafemi Awolowo University Department of Paediatrics and Child Healthmi Awolowo University/ Obafemi Awolowo Universit Off Ede Road Ile-Ife Ile-Ife Osun 220005 Nigeria
REVIEW RETURNED	22-Apr-2024

GENERAL COMMENTS	Dear Authors, Thank you for your response to the comments I raised. My concerns were addressed satisfactorily. Therefore, I recommend the acceptance of your manuscript. Kind regards.
--

REVIEWER	Dr. Andrew Williams Virtual Academic Unit Virtual Academic Unit, Childrens Directorate Virtual Academic Unit, Childrens Directorate Northampton General Hospital Northampton NN1 5BD United Kingdom of Great Britain and Northern Ireland
REVIEW RETURNED	22-Apr-2024

GENERAL COMMENTS	I congratulate the authors of this important paper. I look forward very much to reading further such important papers from them in the future.
--

VERSION 2 – AUTHOR RESPONSE

None